# Advocacy of Precision Allergy Molecular Diagnosis in Decision Making for the Eligibility of Customized Allergen Immunotherapy

Ruperto González-Pérez [1,2,*], Paloma Poza-Guedes [1,2], Fernando Pineda [3] and Inmaculada Sánchez-Machín [1,4]

1   Allergy Department, Hospital Universitario de Canarias, 38320 Tenerife, Spain;
    pozagdes@hotmail.com (P.P.-G.); zerupean67@gmail.com (I.S.-M.)
2   Severe Asthma Unit, Hospital Universitario de Canarias, 38320 Tenerife, Spain
3   Inmunotek SL Laboratories, 28805 Madrid, Spain; fpineda@inmunotek.com
4   Allergen Immunotherapy Unit, Hospital Universitario de Canarias, 38320 Tenerife, Spain
*   Correspondence: glezruperto@gmail.com; Tel.: +34-922-677237

**Abstract:** Allergen immunotherapy (AIT) with aeroallergens is the only disease-modifying treatment for patients with different allergic conditions. Despite the effectiveness of AIT having been proven in both randomized controlled trials and real-world studies, it remains underused in less than 10% of subjects with allergic rhinitis (AR) and/or asthma (A). We aimed to determine the current eligibility for house dust mite (HDM) AIT by means of a precision allergy molecular diagnosis (PAMD@) model in a selected cohort of youngsters with different allergic phenotypes according to the available evidence. A complex response to both HDM and storage mite allergens was depicted regardless of the subjects' basal atopic condition. No solely specific IgE-binding responses to Der p 1, Der p 2, and/or Der p 23 were found in the studied cohort. Despite the patients with A and atopic dermatitis showing significantly higher serum titers to six mite allergens than subjects with AR, no specific molecular profile was regarded as disease specific. Given the increasing complexity of specific IgE responses to the local prevailing aeroallergens, the identification and presence of such molecules are needed in commercially available AIT in the era of precision medicine.

**Keywords:** mites; allergen immunotherapy; molecular allergology; allergens; allergic phenotype

## 1. Introduction

The external exposome plays a key role in the pathobiology of the inflammatory response to aeroallergens, leading to subsequent clinical phenotypes of allergic disease in genetically predisposed individuals. Allergen immunotherapy (AIT) is the only available disease-modifying treatment for respiratory atopic individuals presenting with IgE-mediated allergic rhinitis and/or allergic asthma. Accurate identification of the causative underlying allergen source is mandatory to achieve a successful response to AIT in terms of symptomatic relief and even sustained clinical remission after stopping the treatment [1]. The availability of single allergenic molecules has elicited a revolutionary age in precision allergy molecular diagnosis (PAMD@), including the optimal planning of tailored AIT in polysensitized subjects [2]. In the realm of medicinal product development, children are categorized as a distinct "special population", subject to specific legislation [3]. Regulation (EC) No. 1901/2006 on medicinal products for pediatric use has established a framework of requirements, rewards, and incentives [4]. Its purpose is to guarantee that medicinal products undergo thorough research, development, and authorization processes tailored to meet the therapeutic needs of children. In the context of AIT, the accuracy of patient selection significantly enhances the likelihood of AIT's success [5]. Various patient-dependent factors contribute to this heterogeneity, encompassing sensitization patterns, the efficacy of environmental avoidance measures, the interplay of triggering factors (like infections), microbiome characteristics, epithelial barrier functions, and environmental pollution, as

well as the patient's endotype, phenotype, and associated comorbidities in the context of their active disease [6]. As the natural pathobiology of allergen sensitization across the lifespan has not been fully elucidated and given the prophylactic role of AIT in the so-called "atopic march", the most appropriate timeframe to begin therapy still remains uncertain.

The primary objective of this post-hoc analysis [7] was to comprehensively evaluate the current eligibility status for house dust mite (HDM) AIT within a selected cohort of young individuals who manifest diverse atopic phenotypes. Through a PAMD@ approach, we sought to discern the appropriateness of initiating HDM AIT in this specific group, considering the varied atopic profiles present among these individuals.

## 2. Materials and Methods

### 2.1. Subjects

The current work was restricted to consecutive children and young adults, from 5 to 20 years of age, with an allergist-confirmed diagnosis of active allergic rhinitis (AR), asthma (A), or atopic dermatitis (AD), according to current guidelines [8–10], and enrolled from January of 2021 to January of 2023. Data regarding physician-confirmed food and/or drug allergy were obtained from participants´ past medical records. This investigation was authorized by the local Ethical Committee with reference number P.I.-2017/72, and informed consent was properly collected from all participants and/or parents/guardians for those under 18 years of age. Onset of allergic symptoms after 3 (or more) years of local residency was required to participate in the present investigation. The study workflow included the retrieval of sociodemographic data, past and current medical conditions, and skin prick test (SPT) results to a battery of local inhalants.

### 2.2. Skin Prick Test, Blood Eosinophlis, and Serological Analysis

Percutaneous testing was carried out according to European standards [11], enclosing a diagnostic panel (Inmunotek, Madrid, Spain) with standardized raw extracts (including *Dermatophagoides pteronyssinus* (*D. pteronyssinus*), *Blomia tropicalis* (*B. tropicalis*), *Lepidoglyphus destructor* (*L. destructor*), *Glycyphagus domesticus* (*G. domesticus*), *Tyrophagus putrescentiae* (*T. putrescentiae*), cat and dog dander, grass mix (comprising *Poa pratensis*, *Dactilis glomerata*, *Lolium perenne*, *Phleum pratense*, and *Festuca pratensis*), olives, *Parietaria judaica*, *Artemisa vulgaris*, *Alternaria alternata*, *Aspergillus fumigatus*, *Cladosporium herbarum*, and *Blatella*). Histamine (10 mg/mL) and saline (0.9% NaCl) were used as the positive and negative controls, respectively. The antihistamines were withdrawn a week before the SPT, and wheal diameters were immediately measured after 20 min, with diameters greater than 3 mm being regarded as positive. Blood samples were obtained from all participating individuals, identified with a code label, stored at $-40\,^\circ$C, and thawed immediately prior to the in vitro assay. Six milliliters of blood were transferred to an ethylenediaminetetraacetic acid (EDTA) anticoagulant tube and utilized for a complete blood count (CBC) to quantify the presence of eosinophils in all participants. The blood eosinophil count was assessed at the central laboratory of our institution using a standard automated Beckman Coulter LH755 clinical hematology analyzer [12].

Total IgE and sIgE levels were measured (ALEX2 MacroArray Diagnostics, Vienna, Austria) according to the manufacturer's instructions in all included subjects. In brief, ALEX2 is a multiplex array comprising 295 reagents—including 178 molecules and 117 extracts of airborne allergens and cross-reactive food allergens—with the ability of simultaneously measuring the concentration of serum sIgE (test range: 0.3–50 kUA/L) and total IgE (test range: 1–2500 kU/L). The different allergens and components were coupled onto polystyrene nanobeads; then, the allergen beads were deposited onto a nitrocellulose membrane, as previously published [13]. A total of 17 mite molecular allergens were investigated: Der p 1, Der p 2, Der p 5, Der p 7, Der p 10, Der p 11, Der p 20, Der p 21, Der p23, Der f 1, Der f 2, Blo t 5, Blot 10, Blo t 21, Lep d 2, Gly d 2, and Tyr p 2. Only subjects with a positive SPT to at least one of the corresponding crude mite extracts were included

in this study. Patients under treatment with past or current allergen immunotherapy or biologics were excluded.

### 2.3. Statistical Analysis

Demographic features were summarized through medians and standard deviations for continuous variables and percentages for categorical variables. Kruskal–Wallis, Mann–Whitney U, and Chi-square tests were required for parametric continuous, nonparametric continuous, and categorical variables, respectively. A *p*-value of less than 0.05 was considered statistically significant. All data were analyzed using GraphPad Prism version 8.0.0 for Windows, GraphPad Software, La Jolla, CA, USA.

## 3. Results

### 3.1. Study Population

A total of 73 patients underwent screening, and ultimately, 60 individuals met the eligibility criteria for participation in this study. Within this selected cohort, there were 31 males and 29 females, with an average age of 15.15 years (ranging from 8 to 20 years). Most of the participants in this study were of Caucasian descent (91.3%), predominantly residing in urban areas (72.1%).

These patients were categorized into three distinct groups, AR, A, or AD, based on their current atopic disease and its severity. On a global scale, 60% of the individuals (12 out of 20) experienced severe AR, followed by 55% (11 out of 20) with severe A, and 35% (7 out of 20) with severe AD. All subjects adhered to regular daily treatment protocols, incorporating environmental allergen avoidance measures and standard medical care tailored to their disease stage and severity.

Among these participants, 13 subjects (23.66%) exhibited confirmed food allergies (e.g., milk, egg, and/or tree nuts), while 49 individuals (81.66%) reported a documented family history of atopy. Thirteen patients, comprising six with AR, five with A, and two with AD, were excluded after showing a negative skin prick test (SPT) to local aeroallergens (*n* = 6) or current/former allergen immunotherapy (*n* = 7).

The quantitative analysis of total serum IgE revealed a median value of 380.5 IU/mL (IQR: 999.5) in the investigated population. More specifically, the median total IgE values were 73.15 IU/mL (IQR: 232.02) for AR, 551.5 IU/mL (IQR: 999.5) for A, and 965 IU/mL (IQR: 1962.7) for AD (Table 1).

**Table 1.** Descriptive statistics regarding the basal comorbid conditions and associated clinical features of the studied population (*n* = 60).

|  | **Allergic Rhinitis** | **Allergic Asthma** | **Atopic Dermatitis** |
|---|---|---|---|
| *n* **= 60** | 20 | 20 | 20 |
| **Age (y.o.) median (IQR)** | 14.5 (7) | 16 (5.25) | 15 (5.25) |
| **Sex (F/M)** | 10/10 | 11/9 | 10/10 |
| **Severe atopic disease (%)** | 12/20 (%) | 11/20 (%) | 7/20 (%) |
| **Food allergy (%)** | 3/20 (15%) | 4/20 (20.0%) | 6/20 (30%) |
| **Drug allergy (%)** | 0/20 (0%) | 2/20 (10%) | 1/20 (5%) |
| **SPT+ any aeroallergen (%)** | 20 (100%) | 20 (100%) | 20 (100%) |
| **Total IgE (IU/mL) median (IQR)** | 73.15 (232.02) | 551 (999.5) | 965 (1962.7) |
| **Blood eosinophils/mm$^3$ median (IQR)** | 310 (329.3) | 380 (421.5) | 373 (474.01) |
| **Family history of atopy (%)** | 15/20 (75%) | 16/20 (80%) | 18/20 (90%) |

### 3.2. sIgE Reactivity and Individual Molecular Profiles

Considering individual molecular allergens exclusively, Der f 2 was most frequently identified with sIgE $\geq 0.35$ kU$_A$/L in 54 out of 60 subjects (90%) and Der p 2 in 53 patients (88.3%), followed by Der p 23 (83.3%), Der p 1 and Lep d 2 (81.6%), Gly d 2 (78.3%), Der

f 1 (75%), Der p 5 (70%), Der p 21 (66.6%), Blo t 5 (60%), and Der p 7 (50%) (as shown in Figure 1).

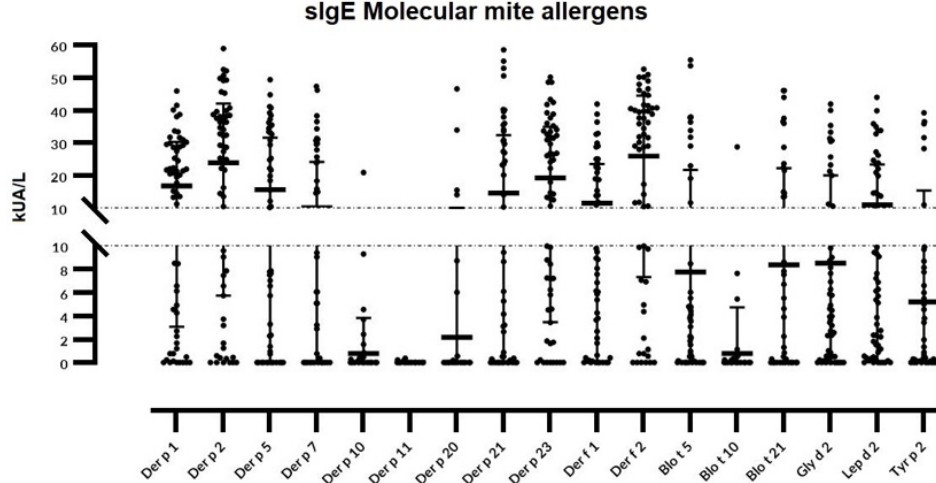

**Figure 1.** Distribution of specific IgE (sIgE) serodominance to a comprehensive panel of 11 molecular mite allergens studied using a microarray in a selected population (*n* = 60) of youngsters (<20 y.o.) with different allergic phenotypes.

The following molecules were found in <50% of the studied population: Tyr p 2 (46.6%), Blo t 21 (45%), Der p 10 (15%), Der p 20 (13.3%), Blo t 10 (11.6%), and Der p 11 (1.6%). Fifty-nine out of sixty (98.3%) individuals depicted sIgE responses to one or more of the investigated molecules. Only two subjects (3.3%) were exclusively sensitized to group 1 or 2 HDM allergens; meanwhile, single reactors were scarcely (5%) found, and limited to Der p 21 and Der p1 in two and one subjects, respectively.

A higher frequency (*p* < 0.05) of sIgE binding was found in patients with A (97.3%) or AD (86.6%) compared to the subjects with AR (49.5%). In addition, patients afflicted with A or AD showed quantitatively higher serum titers (*p* < 0.05) to six mite molecules, namely Der p 2, Der p 5, Der p 7, Der p 21, Der f 1, and Der f 2, than individuals with AR (Table 2).

**Table 2.** Serological analysis of specific IgE (sIgE) responses (kU/L) to 11 house dust mite molecular allergens in patients with allergic rhinitis (AR; *n* = 20), asthma (A; *n* = 20), or atopic dermatitis (AD; *n* = 20). Bold figures indicate quantitatively significant differences (*p* < 0.05) in the mean (SD) sIgE to mite molecular allergens among the three investigated atopic conditions. The % of individuals (*n* = 60) sensitized to the corresponding mite molecular allergen is shown.

| Mite Allergen | Mean sIgE in AR (M ± SD) | % of Sensitized Patients (AR) | Mean sIgE in Asthma (M ± SD) | % of Sensitized Patients (A) | Mean sIgE in Atopic Dermatitis (M ± SD) | % of Sensitized Patients (AD) |
|---|---|---|---|---|---|---|
| Der p 1 | 12.63 ± 0.12 | 75 | 16.69 ± 11.66 | 94.44 | 17.8 ± 0.97 | 83.33 |
| **Der f 1** | **4.36 ± 3.21** | 54.16 | **17.21 ± 9.08** | 94.44 | **15.07 ± 6.69** | 83.33 |
| **Der p 2** | **13.13 ± 9.7** | 79.66 | **33.55 ± 11.66** | 100 | **28.34 ± 18.82** | 88.88 |
| **Der f 2** | **15.39 ± 6.2** | 83.33 | **35.33 ± 11.39** | 100 | **30.35 ± 19.23** | 88.88 |
| **Der p 5** | **7.82 ± 4.91** | 52 | **18.08 ± 7.23** | 88.88 | **24.04 ± 16.84** | 83.33 |
| **Der p 7** | **3.76 ± 2.44** | 22.72 | **12.63 ± 3.35** | 72.22 | **16.31 ± 3.71** | 66.66 |
| Der p 10 | 1.39 ± 0.65 | 18.18 | 0.6 ± 0.27 | 16.16 | 0.4 ± 0.14 | 16.16 |
| Der p 11 | <0.35 | 0 | <0.35 | 0 | 0.36 ± 0.03 | 5.55 |
| Der p 20 | 2.06 ± 1.94 | 16.66 | 0.43 ± 0.31 | 16.66 | 3.85± 1.23 | 16.66 |
| **Der p 21** | **7.29 ± 3.64** | 58.33 | **16.46 ± 7.23** | 77.77 | **24.53 ± 4.32** | 77.77 |
| Der p 23 | 14.61 ± 9.12 | 70.83 | 20.40 ± 3,28 | 94.44 | 24.07 ± 8.01 | 88.88 |

The aggregation of molecules beyond group 1 or 2 HDM allergens, such as Der p 5, Der p 7, Der p 21, Der p 23, Blo t 5, Lep d 2, and Gly d 2, confirmed a marked pleiomorphic molecular response in youngsters subjected to a high exposure of both HDMs and SMs

and influenced by perennial subtropical climate conditions. The A and AD phenotypes both showed a more complex aggregation pattern of molecules, including concurrent sensitization to ≥eight HDM mite allergens, in contrast to most of the subjects (>83%) with AR displaying a concomitant sIgE response to <eight HDM molecules (Table 3).

**Table 3.** Number of identified house dust mite (HDM) molecular allergens and corresponding basal atopic disease (i.e., allergic rhinitis, asthma, and atopic dermatitis) in 60 patients studied via a microarray. Eleven HDM mite molecular allergens were investigated: Der p 1, Der p 2, Der p 5, Der p 7, Der p 10, Der p 11, Der p 20, Der p 21, Der p23, Der f 1, and Der f 2. Most of the subjects (83.34%) with allergic rhinitis displayed a specific IgE response <eight mite HDM molecules, in contrast to patients with asthma (66.66%) or atopic dermatitis (66.66%), who showed a polysensitization profile to ≥eight individual HDM allergens.

| Number of Identified HDM Allergens | Allergic Rhinitis (*n* = 20) | Asthma (*n* = 20) | Atopic Dermatitis (*n* = 20) |
|---|---|---|---|
| 0 | 0 | 0 | 1 |
| 1 | 3 | 0 | 1 |
| 2 | 3 | 0 | 0 |
| 3 | 1 | 1 | 0 |
| 4 | 3 | 0 | 0 |
| 5 | 6 | 1 | 0 |
| 6 | 1 | 2 | 3 |
| 7 | 3 | 1 | 2 |
| 8 | 3 | 11 | 9 |
| 9 | 0 | 1 | 3 |
| 10 | 1 | 1 | 0 |

## 4. Discussion

Allergies emerge as consequences of intricate gene–environment interactions, marked by their diverse nature, varying clinical phenotypes, and the accompanying processes of inflammation. Our findings indicated that IgE reactivity profiles in children to multiple mite allergen components increase in their concentration and complexity depending on their basal atopic disease. The current commercial availability of additional mite molecular allergens has expanded our diagnostic possibilities beyond the dichotomy of group 1 or 2 HDM allergen sensitization, a limitation that has been effectively addressed in the studied population through the implementation of PAMD@. In fact, 9, 11, and 13 mite molecules showed a corresponding sIgE-binding frequency ≥50% in patients afflicted with AR, AD, and A, respectively. Former longitudinal investigations, confirming the molecular spreading hypothesis, have elegantly showed that only children with concurrent sIgE responses to groups 1 and 2 mite allergens had the highest risk of asthma and significantly increased exhaled NO [14,15]. Interestingly, no significant ($p = 0.36$) differences in the frequency of sensitization to both group 1 and 2 HDM allergens were found in our studied cohort, regardless of their basal atopic condition.

In line with previous research, Der p 5 and Der p 21, two allergens with no relevant IgE cross-reactivity (despite showing a similar three-dimensional crystallographic helical-in-nature structure) [16,17], were highly (>65%) identified in the present population. In fact, the structure of Der p 5 suggests that a Der p 5 dimer may also have a propensity to bind hydrophobic compounds, shifting the immune response from tolerance to Th2-type immune responses that are associated with allergic inflammation [18–20]. Although both allergens have been described as clinically relevant molecules, especially in asthmatic subjects, a significant IgE-binding frequency was also found for the AD subjects in contrast to the patients solely afflicted with AR [21,22].

Recently, Der p 23, a peritrophin-like protein of 8 kD that synthesizes the intestinal tract of mites, has been also strongly associated with asthma in different European cohorts [23,24]. Despite Der p 23 exhibiting a high level of allergenic activity, binding IgE in around 70% of mite-allergic subjects [25] and with Der p 23 monomolecularly sensitizing subjects

without sIgE to other HDM allergens having been reported, no single reactors (0%) to Der p 23 were identified in the present cohort. In fact, sensitization to Der p 23 was found equally relevant in terms of prevalence and sIgE quantification across the three investigated atopic phenotypes. In addition, our study highlighted an increased sensitization rate (50%) for patients with AD and A to Der p 7, which has been biologically described as a potent allergenic molecule activity, like Der p 5, Der p 21, and/or Der p 23, in specific cohorts [26,27].

Although most of the subjects included in the present investigation (95%) were polysensitized, no management strategies to AIT have been standardized for such patients as of yet [28]. According to previous reports, the clinical efficacy of AIT in inducing protective IgG was only achieved in those patients sensitized to Der p 1 and/or Der p 2, and to a lesser extent to Der p 23, but not to other relevant allergens, namely Der p 5, Der p 7, and/or Der p 21 [29,30]. Conforming to this evidence, most of the studied subjects in our population may not benefit from AIT, as following these criteria, only 8.3% of the selected individuals would be eligible for PAMD@-driven AIT, regardless of their basal atopic phenotype. Conversely, as recent research has stated that sIgG4 to HDM components do not qualify as a biomarker to evaluate the efficacy of AIT [31], additional research, including the assessment of the allergenic activity and clinical impact of individual allergens, is warranted to identify further prognostic markers for AIT efficacy in certain populations. In fact, former studies have highlighted the usefulness of the basophil activation test (BAT) dose–response curves to appraise the clinical efficacy of hymenoptera venom AIT [32,33].

Another intriguing issue, and especially in pediatric patients, is when should AIT be started. Current guidelines recommend that AIT can be initiated at 5 years of age [34]; meanwhile, allergen sensitization can start as early as 12 months of age [35]. In this regard, a comprehensive individual patient selection process, including the sensitization profile related to the current allergen exposome, the active underlying atopic disease, and associated comorbidities, increases the adherence to prescribed medications and the success of AIT [36]. Despite previous studies having confirmed that PAMD@ changed the choice of relevant allergens for allergen-specific immunotherapy in at least 50% of cases [37,38], further investigations are warranted to investigate the prophylactic role of AIT in these patients with a clinically relevant sensitization under the age of 5 years old before the development of complete longitudinal mite sensitization trajectories.

A few limitations should be noted. This post-hoc study, which solely focused on children and youngsters, has a relatively restricted sample size, and the reported data came from a single center. Additionally, two out of sixty individuals (3.33%) with a positive SPT to local aeroallergens—one asthmatic subject and one subject with atopic dermatitis—could not be identified using the implemented multiplex array.

As planned, AIT is grounded in the fundamental assumption that patients primarily exhibit sensitization to major allergens within the allergenic source rather than minor allergens [39]. Two considerations may be addressed in this regard: Firstly, contemporary research strongly advocates for a departure from the conventional practice of classifying allergens solely based on their IgE-binding frequency. Recent studies have proposed a more nuanced approach, urging the evaluation of allergen clinical relevance in diverse geographical scenarios [40,41]. This shift in perspective encourages a more tailored and context-sensitive identification of allergens within AIT protocols. Secondly, rather than adhering to the traditional restriction of AIT to patients solely sensitized to the 'major allergens', a more progressive strategy involves the identification and incorporation of 'minor allergens' into commercially available AIT preparations [42]. By implementing this strategy, the therapeutic landscape broadens, offering potential benefits to a growing demographic of polysensitized individuals. This inclusive approach recognizes the intricate nature of allergic responses, acknowledging that patients may present with sensitivities to a combination of major and minor allergens.

In conclusion, the large majority of youngsters afflicted with various respiratory or skin allergic conditions, as investigated in the present cohort, were polysensitized

to different mite molecules with proven allergenic activity. This complexity makes the prescription of personalized AIT challenging. As climate change continues to play a pivotal role in the production, allergenicity, and concentration of airborne allergens, a child-centered AIT undoubtedly requires innovative approaches. This includes conducting longitudinal and prospective clinical trials to provide evidence for safer and more effective immunomodulatory interventions during the early stages of these atopic diseases.

**Author Contributions:** Conceptualization, R.G.-P., P.P.-G. and F.P.; methodology, R.G.-P., P.P.-G. and F.P.; software, F.P.; validation and formal analysis, I.S.-M. and F.P.; investigation, I.S.-M., R.G.-P., P.P.-G. and F.P.; resources, I.S.-M. and F.P.; data curation, R.G.-P., P.P.-G. and F.P.; writing—original draft preparation, R.G.-P. and P.P.-G.; writing—review and editing, R.G.-P., P.P.-G. and F.P.; project administration R.G.-P., P.P.-G. and I.S.-M.; funding acquisition R.G.-P. and P.P.-G. All authors have read and agreed to the published version of the manuscript.

**Funding:** This study was funded by the Fundación Canaria Instituto de Investigación Sanitaria de Canarias (FIISC), Servicio Canario de Salud, grant number OA17/042.

**Institutional Review Board Statement:** This study was conducted according to the guidelines of the Declaration of Helsinki and approved by the Institutional Ethics Committee of CEIC Hospital Universitario de Canarias, Tenerife, Spain, with reference number P.I.-2017/72, on 30 October 2017.

**Informed Consent Statement:** Informed consent was obtained from all subjects involved in this study.

**Data Availability Statement:** The data that support the findings of this study are available from Servicio Canario de Salud, but restrictions apply to the availability of these data, which were used under license for the current study and thus are not publicly available. These data are, however, available from the authors upon reasonable request and with the permission from the Servicio Canario de Salud.

**Conflicts of Interest:** The authors declare no conflict of interest. The funders had no role in the design of the study; in the collection, analyses, or interpretation of data; in the writing of the manuscript, or in the decision to publish the results.

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
