# Peer review of "Advocacy of Precision Allergy Molecular Diagnosis in Decision Making for the Eligibility of Customized Allergen Immunotherapy"

_cimb, doi:10.3390/cimb45120623_

Round 1
Reviewer 1 Report
Comments and Suggestions for Authors
The manuscript is interesting and adds data to the evaluation of the theoretical and practical importance of epitope spreading and the possibilities that have opened up with multiplex methods.
There are some aspects to improve:
“…22 out of 20 (60%) individuals suffered from severe AR, followed by 55% (11 out 20) with severe A,and 35% (7 out of 20)…”
This chapter should be written better, as it can only be understood by examining the table I
“….Conversely, as recent research state that sIgG4 to HDM components do not qualify as a biomarker to evaluate the efficacy of AIT [22], additional research including the assessment of the allergenic activity and clinical impact of individual allergens is warranted to identify further prognostic markers for AIT efficacy in certain populations….”
Several studies have highlighted how the BAT dose-response curves are useful for evaluating the effectiveness of AIT for hymenoptera venom, it could be a research suggestion to include.
“…Additionally, two out of 60 individuals (3.33%) with a positive skin prick test (SPT) to local aeroallergens one asthmatic and one subject with atopic dermatitis could not be identified by the implemented multiplex array..”.
The multiplex array is not as sensitive as conventional methods, have patients been studied with the ImmunoCap?
Author Response
Authors´ Response to Reviewers [cimb-2755710]:
Rev 1
Comments and Suggestions for Authors
The manuscript is interesting and adds data to the evaluation of the theoretical and practical importance of epitope spreading and the possibilities that have opened up with multiplex methods.
There are some aspects to improve:
“…22 out of 20 (60%) individuals suffered from severe AR, followed by 55% (11 out 20) with severe A,and 35% (7 out of 20)…”
This chapter should be written better, as it can only be understood by examining the table I
Answer: Thank you for the comment, the text has been modified to improve clarity as follows: “A total of 73 patients underwent screening, and ultimately, 60 individuals met the eligibility criteria for participation in the study. Within this selected cohort, there were 31 males and 29 females, with an average age of 15.15 years (ranging from 8 to 20). The majority of participants in this study were of Caucasian descent (91.3%), residing predominantly in urban areas (72.1%). Patients were categorized into three distinct groups—AR, A, or AD—based on their current atopic disease and its severity. On a global scale, 60% of the individuals (12 out of 20) experienced severe AR, followed by 55% (11 out of 20) with severe A, and 35% (7 out of 20) with severe AD. All subjects adhered to regular daily treatment protocols, incorporating environmental allergen avoidance measures and standard medical care tailored to their disease stage and severity. Among the participants, 13 subjects (23.66%) exhibited confirmed food allergies (milk, egg, and/or tree nuts), while 49 individuals (81.66%) reported a documented family history of atopy. Thirteen patients—comprising 6 with AR, 5 with A, and 2 with AD—were excluded after showing a negative skin prick test (SPT) to local aeroallergens (n = 6) or current/former allergen immunotherapy (n = 7). The quantitative analysis of total serum IgE revealed a median value of 380.5 IU/mL (IQR: 999.5) in the investigated population. More specifically, the median total IgE values were 73.15 IU/mL (IQR: 232.02) for AR, 551.5 IU/mL (IQR: 999.5) for A, and 965 IU/mL (IQR: 1962.7) for AD (Table 1).
“….Conversely, as recent research state that sIgG4 to HDM components do not qualify as a biomarker to evaluate the efficacy of AIT [22], additional research including the assessment of the allergenic activity and clinical impact of individual allergens is warranted to identify further prognostic markers for AIT efficacy in certain populations….”
Several studies have highlighted how the BAT dose-response curves are useful for evaluating the effectiveness of AIT for hymenoptera venom, it could be a research suggestion to include.
Answer: Thank you for the accurate comment, the following sentence and corresponding references have been added: “In fact, former studies have highlighted the usefulness of basophil activation test (BAT) dose-response curves to appraise the clinical efficacy of hymenoptera venom AIT [Eržen R, et al. Allergy 2012, Eberlein B. Front Immunol 2020].”
“…Additionally, two out of 60 individuals (3.33%) with a positive skin prick test (SPT) to local aeroallergens one asthmatic and one subject with atopic dermatitis could not be identified by the implemented multiplex array..”.
The multiplex array is not as sensitive as conventional methods, have patients been studied with the ImmunoCap?
Answer: Thank you for the precise comment, regrettably the current population has not been studied with the ImmunoCap assay. The Authors will definitely consider this relevant suggestion of interest for further research in order to improve the overall sensitivity of the study protocol.
Reviewer 2 Report
Comments and Suggestions for Authors
It is interesting that many subjects who participated in the study were sensitized to multiple allergens in addition to the main allergen, and that the degree of sensitization varied depending on the disease. Currently, AIT can only administer major allergens, but it would be ideal if the allergen to which each patient is sensitized could be administered. I hope for the development of research toward clinical application.
It's a very subtle point, but I think that P@MD on lines 36 and 191 is an error in PAMD@.
Author Response
Authors´ Response to Reviewers [cimb-2755710]:
Rev 2
It is interesting that many subjects who participated in the study were sensitized to multiple allergens in addition to the main allergen, and that the degree of sensitization varied depending on the disease. Currently, AIT can only administer major allergens, but it would be ideal if the allergen to which each patient is sensitized could be administered. I hope for the development of research toward clinical application.
It's a very subtle point, but I think that P@MD on lines 36 and 191 is an error in PAMD@.
Answer: Thank you for the precise remark, the text has been modified as suggested in both lines 36 and 191.
Reviewer 3 Report
Comments and Suggestions for Authors
The work of González-Pérez et al. [cimb-2755710] is interesting and contains an interesting approach to the topic of dust mite allergy. It needs some minor corrections though.
1. In the section of the introduction a broader problem should be shown. Also aim should be clearly formulated.
2. Line 50: Please provide the number of approval of the Ethical Committee.
3. In demographic data analyses the median and quartiles rather than mean and SD should be provided.
4. Line 168- IgE cross-reactivity despite showing a similar three-dimensional structure: What did the author mean about a similar 3D structure? Was It based on aa sequence and folding? 1st dimension structure needs about 70% homology to consider it as possibly cross-reactive.
5. The introduction and discussion seem to be not discursive enough.
In general, the work seems interesting. After that minor corrections can be published.
Comments on the Quality of English LanguageThe manuscript is pretty well written. It needs a few minor corrections but in general, it is ok. The English needs also some improvement but is is ok.
Author Response
Authors´ Response to Reviewers [cimb-2755710]:
Rev 3
The work of González-Pérez et al. [cimb-2755710] is interesting and contains an interesting approach to the topic of dust mite allergy. It needs some minor corrections though.
- In the section of the introduction a broader problem should be shown. Also aim should be clearly formulated.
Answer: Thank you for the precise comment. The following modifications have been enclosed in the introduction:
“In the context of AIT, the accuracy of patient selection significantly enhances the likelihood of AIT success [Canonica GW, et al. World Allergy Organ J 2015]. Various patient-dependent factors contribute to this heterogeneity, encompassing sensitization patterns, the efficacy of environmental avoidance measures, the interplay of triggering factors like infections, microbiome characteristics, epithelial barrier functions, environmental pollution, as well as the patient's endotype, phenotype, and associated comorbidities in the context of their active disease [Alvaro-Lozano M, et al. PAI 2020].”
“In the realm of medicinal product development, children are categorized as a distinct "special population," subject to specific legislation [Mahler V, et al. Clin Transl Allergy 2020]. Regulation (EC) No 1901/2006 on medicinal products for pediatric use establishes a framework of requirements, rewards, and incentives [Bonertz A, et al. Allergy 2018]. Its purpose is to guarantee that medicinal products undergo thorough research, development, and authorization processes tailored to meet the therapeutic needs of children.”
The aim has also been reformulated as follows: “The primary objective of this post-hoc analysis is to comprehensively evaluate the current eligibility status for house dust mite (HDM) allergen immunotherapy (AIT) within a selected cohort of young individuals who manifest diverse atopic phenotypes [3]. Through a PAMD@ approach, we seek to discern the appropriateness of initiating HDM AIT in this specific group, considering the varied atopic profiles present among these individuals.”
- Line 50: Please provide the number of approval of the Ethical Committee.
Answer: Thank you for the comment, the correspondent number of approval has been provided.
- In demographic data analyses the median and quartiles rather than mean and SD should be provided.
Answer: Thank you for the comment. Modifications in accordance have been made in table #1.
- Line 168- IgE cross-reactivity despite showing a similar three-dimensional structure: What did the author mean about a similar 3D structure? Was It based on aa sequence and folding? 1st dimension structure needs about 70% homology to consider it as possibly cross-reactive.
Answer: Thank you for accurate remark, the following information has been provided to address this point:
Line 167 of the marked copy of the manuscript: “In line with previous research, Der p 5 and der p 21, two allergens with no relevant IgE cross-reactivity despite showing a similar three-dimensional crystallographic helical in nature structure [Weghofer M, et al. Int. Arch. Allergy Immunol 2008, Weghofer M, et al. Allergy 2008], were highly (>65%) identified in the present population. In fact, the structure of Der p 5 suggests that a Der p 5 dimer may also have a propensity to bind hydrophobic compounds shifting the immune response from tolerance to Th2-type immune responses that are associated with allergic inflammation [Trompette A, et al Nature 2009, Thomas WR, et al. Curr Allergy Asthma Rep. 2005].”
- The introduction and discussion seem to be not discursive enough.
Answer: Thank you for the comment. Besides the changes provided in the introduction (please, see above answer #1) the following texts has been added in the discussion section:
“Allergies emerge as consequences of intricate gene-environment interactions, marked by their diverse nature, varying clinical phenotypes, and the accompanying processes of inflammation.”
“The current commercial availability of additional mite molecular allergens has expanded our diagnostic possibilities beyond the dichotomy of group 1 or 2 HDM allergen sensitization, a limitation effectively addressed in the studied population through the implementation of PAMD@.”
“In line with previous research, Der p 5 and der p 21, two allergens with no relevant IgE cross-reactivity despite showing a similar three-dimensional crystallographic helical in nature structure [Weghofer M, et al. Int. Arch. Allergy Immunol 2008, Weghofer M, et al. Allergy 2008], were highly (>65%) identified in the present population. In fact, the structure of Der p 5 suggests that a Der p 5 dimer may also have a propensity to bind hydrophobic compounds shifting the immune response from tolerance to Th2-type immune responses that are associated with allergic inflammation [Trompette A, et al Nature 2009, Thomas WR, et al. Curr Allergy Asthma Rep. 2005].”
“In fact, former studies have highlighted the usefulness of basophil activation test (BAT) dose-response curves to appraise the clinical efficacy of hymenoptera venom AIT [Eržen R, et al. Allergy 2012, Eberlein B. Front Immunol 2020].”
“Despite previous research has confirmed that PAMD@ changed the choice of relevant allergens for allergen-specific immunotherapy in at least 50% of cases [Izmailovich M, et al. Cells 2023],…”
In general, the work seems interesting. After that minor corrections can be published.